# Focus Group Discussions on Food Waste: An Empirical Application Providing Insights into Rural and Urban Households in Greece

Vasiliki Aitsidou [1], Evangelia Michailidou [1], Efstratios Loizou [2], Georgios Tsantopoulos [3] and Anastasios Michailidis [1,*]

1   Department of Agricultural Economics, School of Agriculture, Aristotle University of Thessaloniki, 54124 Thessaloniki, Greece; vaitsido89@gmail.com (V.A.); michevan@plandevel.auth.gr (E.M.)
2   Department of Regional and Cross Border Development, University of Western Macedonia, 50100 Kozani, Greece; eloizou@uowm.gr
3   Department of Forestry and Management of the Environment and Natural Resources, Faculty of Agricultural and Forestry Sciences, Democritus University of Thrace, 68200 Orestiada, Greece; tsantopo@fmenr.duth.gr
*   Correspondence: tassosm@auth.gr

**Abstract:** This paper demonstrates the utility of the focus group discussions (FGDs) methodology in the scientific exploration of food waste. The main objective is to show how FGDs can be designed and implemented by collecting data on household food waste (HFW). The paper provides an empirical application of FGDs to members of urban and rural households in Greece through 10 steps. It is qualitative research that was implemented as a supplement in the framework of a large-scale study on HFW, providing an in-depth interpretation of the statistical results that were arrived at. The research shows that FGDs are an effective data collection methodology that reveals insights into HFW through interactions and complex behaviors. Further, the methodology used gives the opportunity to bring information to the fore. The role of women in relation to food-related responsibilities in the Greek household and the impact of rural experiences on HFW composition constitute two topics under exploration. A detailed understanding of HFW examined through the FGDs methodology enriches the global bibliography, mainly for the case of Greece. In addition, useful information is provided to local and governmental bodies, enabling them to collaborate with academics and experts in food waste management. There is a willingness among household members to raise their awareness of HFW reduction and prevention.

**Keywords:** factors; household food waste; interviews; qualitative research; rurality

## 1. Introduction

Food wastage is a complex global phenomenon that entails a variety of social, environmental, and economic issues [1] and occurs at different stages along the food supply chain (FSC) [2]. Globally, one-third of edible food is wasted every year [3], while millions of people in developing countries are undernourished [4]. Food waste is mainly observed at the "public and household consumption" stage [1,5], but differs across countries and local communities—urban and rural. [6,7].

HFW (household food waste) composition in Greece does not differ significantly from other Mediterranean countries [8–10]). The most rejected foods in Greek households are fresh fruits, vegetables [10,11], and cooked-food leftovers, not excluding meat or fish [10]. HFW composition is strictly connected to the available knowledge, formed perceptions, and attitudes formed through food-related household activities, mainly including shopping, cooking, and leftovers management [11–13]. More specifically, people buy and cook more food than they should. In this way, they produce food waste which they are not going to consume in other meals or in feeding animals. Moreover, this situation is further burdened

by the ignorance of food labels content. The most common issue that increases HFW is that people often struggle to interpret the dates on food labels [14].

A plethora of studies have been conducted globally in recent years exploring HFW, the majority of which suggest that specific factors influence HFW composition [15]. The most prominent factors are age, gender, profession, and place of residence [15–18]. Based on the latest literature, older individuals [19], males [20], those who are unemployed or seeking employment [21], as well as rural inhabitants [22] tend to waste less food compared to their younger, female, employed, and urban counterparts. Nevertheless, there is a lack of evidence substantiating these observations [13].

The main objective of the current paper is to advance the understanding of factors influencing HFW, and, thus, to shed light on its causes and consequences. More specifically, this paper concerns supplementary research that was included [23] within a comprehensive empirical study on HFW in Greece. The need to interpret some of the surveys' statistical results induced the research team to organize and implement in-depth interviews. Within this framework, the FGD$_S$ method [24] was used. The selection of this method compelled the research team to expound upon and enrich the global bibliography with detailed insights into the most common quantitative findings (factors) concerning HFW [9,20,21].

This research addresses seven crucial findings regarding HFW composition, providing a clearer insights into them. The following are the key findings presented in detail:

1.  Women are the primarily contributors to the composition of HFW. According to the descriptive statistics, 62% of women are responsible for household food purchases and 87% of them for cooking;
2.  Individuals who are 26–45 years old are more likely to waste more food than the elderly (65+ years old). Based on the descriptive statistics, 70% of individuals aged 65 and over declared that they never discard food in their households while 65% of individuals aged 18–25 are more likely to compose HFW;
3.  Younger people (18–25 years old) are more likely to waste more food than the 26–45 age range The descriptive statistics reveal that 26% of individuals aged 18–25 discard more food than 17% of individuals aged 26–45 who discard less food at the household level;
4.  Rural food-related experiences have a positive influence on HFW composition. The two-step cluster analysis shows that household members who have lived in the past in a rural area (75%) and/or come from a rural area (80%) are characterized by socio-ecological awareness of food waste issues;
5.  Food labeling dates have a negative influence on HFW composition. The results of the categorical principal components analysis (CPCA) disclose that household members are not used to consuming foods after the "best before date" (median 2.68) or to buying and/or consuming food close to their expiry date (median 1.95);
6.  Consumers' attitudes encompass both beneficial and non-harmful food-related habits and behaviors with regard to HFW composition, e.g., to use the FIFO consumption method (first-in-first-out), to reduce overconsumption, etc. The most relevant result with regard to the above attitude is that the CPCA analysis reveals that household members use the FIFO method (median 2.85), eat leftovers in other meals (median 3.97), eat more than they need (median 2.80), and try to discard less food (median 4.50);
7.  There are insufficient or erroneous perceptions of environmental issues linked to HFW composition such as food miles and climate change. Based on the CPCA results, household members are not aware of the impact of human activity on nature and climate change as well (median 2.48), and they do not realize that local foods are less harmful to the environment than imported foods (median 2.55).

The current paper presents a first attempt, for the case of Greece, to interpret the most common issues related to HFW composition. The innovation of the paper concerns the application of the FGDs qualitative methodology on food waste issues. A step-by-step FGDs process is presented, providing a better understanding of food waste production by household members. More specifically, the short daily food stories of FGDs participants

are proven to be a vital source of information that explains the composition of food waste in rural and urban households in the study area. Policies and limitations are also included at the end of the paper.

## 2. Methods

Data for this research were collected from a sample of the Greek population in the municipality of Eordaia, Western Macedonia. Firstly, an empirical study on HFW was conducted through a survey questionnaire, and 279 valid responses were collected. The stratified sampling technique [5] indicated a sample of 193 households from the 1 urban community and 86 households from the 25 rural communities that comprise the study area. The data were collected, encoded, recorded, and analyzed statistically. Data analysis revealed findings on food waste issues but proved insufficient to provide clear evidence on HFW causes and consequences. Therefore, it was deemed essential to conduct supplementary qualitative research in order to identify why and how specific demographics make an impact on HFW composition.

The FGDs methodology [25] was selected, and the necessary data were compiled in 10 steps, which are described below (Figure 1) [26]:

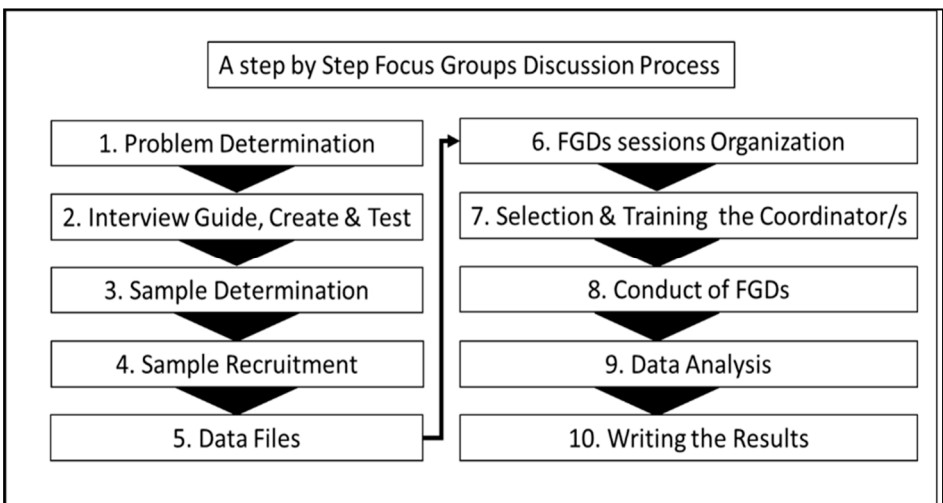

**Figure 1.** FGD$_S$ methodology, a step-by-step approach to HFW issues.

Step 1—Problem determination: According to the statistical analysis of a large-scale study on HFW, some results with no clear understanding were obtained. In order to provide clear evidence relating to factors, causes, and consequences affecting HFW, the FGDs method was chosen to provide valuable insights.

Step 2—Interview guide, creation, and test: An interview guide of 17 open-ended questions (Appendix A, Table A1) was created based on the statistical results mentioned above (step 1). The interview guide was tested to ensure that all the participants understood the questions, providing the required information. In this manner, the framework for the guide's validity and reliability was established [27].

Step 3—Sample determination: Participants were selected according to the purposive sampling technique [28,29], whereby they met a combination of heterogeneous and homogeneous criteria. The heterogeneity of the sample was attributed to the participants' different genders, ages, educational levels, and residential places—urban or rural. Their homogeneity lay in the fact that all of them were residents of the same municipality. Furthermore, the heterogeneity promotes motivation for discussion, while the homogeneity facilitates the communication among interviewees, thus encouraging them to share their perceptions, ideas, and experiences [30].

Step 4—Sample recruitment: Interviewees participated voluntarily without any recruitment involving monetary incentives. This step is not a prerequisite for implementing

FGDs, as these payments are not a common motivation and are mainly used to increase interview participation [31].

Step 5—Data files: All interviewees, before the session started, received a specially structured folder that included four files:

i.     Information sheet which included all the details about the research process, the assurance of their anonymity, as well as contact information.

ii.    Answer sheet (Appendix A, Table A2). The interviewees were asked to briefly record their opinions after the discussion of each question. The need to record the answers of the interviewees themselves was created in order to avoid losing answers. The coordinator was simultaneously briefly recording the answers, along with the interactions among the interviewees [32].

iii.   Practical advice sheet which included 9 tips (Appendix A, Figure A1) about how to reduce and/or prevent HFW, published by the Food and Agriculture Organization of the United Nations [33].

iv.    "Every Food on the Right Shelf" sheet which included information about the proper food storage places on household shelves, in kitchens, and in refrigerators, published by the Academy of Nutrition and Dietetics [34].

Both the third and fourth sheets were given to interviewees at the end of their session as a gift—reminder. The main purpose of this gift was to encourage all interviewees to enhance HFW prevention and/or reduction.

Step 6—FGDs sessions organization: An adequate number of FGDs sessions is 2–6 [31,35], and an adequate group size is 4–12 participants [36,37] in each one. In the present research, 13 individuals were invited to one of the two FGDs sessions held—$FGD_{sA}$ and $FGD_{sB}$. A total of 6 and 7 individuals participated in the two groups A and B, respectively (Tables 1 and 2).

**Table 1.** Focus Group A—heterogeneous participants' characteristics.

| Interviewee Number | Gender | Age (Age Group) | Educational Level | Residence |
|:---:|:---:|:---:|:---:|:---:|
| 1 | female | 18 (18–25) | Secondary | urban |
| 2 | male | 26 (26–35) | Post-secondary | urban |
| 3 | female | 53 (46–55) | Secondary | urban |
| 4 | male | 59 (56–65) | Post-secondary | urban |
| 5 | female | 57 (56–65) | Primary | rural |
| 6 | male | 67 (65+) | Tertiary | urban |

**Table 2.** Focus Group B—heterogeneous participants' characteristics.

| Interviewee Number | Gender | Age (Age Group) | Educational Level | Residence |
|:---:|:---:|:---:|:---:|:---:|
| 1 | female | 18 (18–25) | Secondary | urban |
| 2 | female | 55 (46–55) | Secondary | urban |
| 3 | female | 55 (46–55) | Primary | urban |
| 4 | female | 59 (56–65) | Primary | rural |
| 5 | male | 60 (56–65) | Tertiary | urban |
| 6 | male | 62 (56–65) | Tertiary | urban |
| 7 | female | 66 (65+) | Primary | rural |

Step 7—Selecting and training the coordinator/s: The main and sole coordinator in both FGDs sessions was the researcher—the interviewer—who informed the participants—the interviewees—on the specific issues below [31]:

●     There are no right or wrong answers;
●     If someone does not understand the question, it will be rephrased;

- Everyone, in turn, will take the floor expressing their personal opinion;
- It will be acceptable to change their opinion after the commentary discussion of each question;
- Comments and answers will be recorded by hand, both by the interviewer and the interviewees;
- The necessity to obtain an oral consent from each participant.

Step 8—Conduct of $FGD_S$: The two FGDs were conducted over 2 days and during the afternoon hours, as indicated by the bibliography. $FGD_{SA}$ lasted a total of 2 h and 10 min (17:30 p.m. to 19:40 p.m.), and $FGD_{SB}$ lasted a total of 2 h (17:30 p.m. to 19:30 p.m.). A special venue, one of the oldest in the research area, was chosen. The most important criterion in selecting a dedicated space was the maintenance of quietness during the whole interview [27].

Step 9—Data analysis: Data were recorded, coded, and checked, firstly by group and afterward all together [33]. This process was conducted by experts (academics and coordinator) in qualitative research, as required [25]. It is worth noting that the data analysis was based only on the handwritten files mentioned above (steps 4 and 6).

Step 10—Writing the results: For the purpose of this paper, the results will be presented in the Section 3. However, the FGDs results, as part of the large-scale research on HFW carried out, are integrated into the Section 4, in order to provide a better understanding of the statistical results.

## 3. Results

Thirteen local inhabitants participated in two FGDs. Eight participants were women and five men. Their age ranged from 18 to 67 years. As regards education, four participants were primary school graduates (basic education level), four were high school and post-secondary school graduates (high education), and five were university graduates (higher education). The majority of them (10 participants) lived in an urban center, and only 3 lived in a rural community. However, all of them had lived (in the past) in a rural area for a long time. Hence, they are characterized by rural past and rural memories. The demographic characteristics of the FGDs participants are presented in Table 3.

**Table 3.** Characteristics of FGDs participants.

| Focus Group | | Gender | Age (Age Group) | Education Level | Residence Area | Rural Past |
|---|---|---|---|---|---|---|
| $FGDs_A$ | 1. | female | 18 (18–25) | high | urban | yes |
| | 2. | male | 26 (26–35) | higher | urban | yes |
| | 3. | female | 53 (46–55) | high | urban | yes |
| | 4. | male | 59 (56–65) | higher | urban | yes |
| | 5. | female | 57 (56–65) | basic | rural | yes |
| | 6. | male | 67 (65+) | higher | urban | yes |
| $FGDs_B$ | 1. | female | 18 (18–25) | high | urban | yes |
| | 2. | female | 55 (46–55) | high | urban | yes |
| | 3. | female | 55 (46–55) | basic | urban | yes |
| | 4. | female | 59 (56–65) | basic | rural | yes |
| | 5. | male | 60 (56–65) | higher | urban | yes |
| | 6. | male | 62 (56–65) | higher | urban | yes |
| | 7. | female | 66 (65+) | basic | rural | yes |

### 3.1. How Demographics Affect HFW

3.1.1. Gender

HFW composition is strictly connected to specific household members. The majority of the participants declared that women are responsible for all the food-related household activities and, consequently, for HFW. Food purchases, cooking, and food storage on household shelves (kitchen, fridge, etc.) are activities that mainly women undertake in

most households in the research area. Nevertheless, a woman is more likely to reheat and eat food leftovers in other meals, seeking out best practices on HFW management, e.g., food and vegetable preservation in the freezer. Furthermore, women are more likely to adopt healthy eating habits as they are interested, more than anyone else, in their appearance. The most noteworthy responses are listed below:

"... my wife cleans the table and does the dishes after each meal ... women do most of the housework ..."

"... I help my mother with food purchases! I drive her to the supermarket twice a month ..." (*laughs*)

"... I am trying to eat healthy ... I am getting older ... it is hard for me to remain slim and healthy."

"... sometimes my children prefer ordering take out to eating the cooked food ..."

"... cooked food is not always consumed on time ... my husband does not like to eat reheated food ..."

### 3.1.2. Age

The majority of responses indicated that young people are more likely to generate HFW. Their daily routine does not include any household activities, e.g., consumption of leftovers, checking of food labels and food dates, etc. On the contrary, the older age group is more responsible and mature enough to perceive that food waste implies household income in the rubbish. In fact, age differences, especially between young and elderly people, concerning HFW were directly attributed to living conditions (in the past) and experiences. This is evidenced by the following responses:

"... older people lived under difficult circumstances, like wars when they were young ... these people appreciate food more than the young nowadays ..."

"... even in towns, there were not food markets like now ... we can access food almost 24 h a day ..."

"... older people, when they were young had to find, exchange or produce their own food and they also had to manage it in the best way possible ..." (*participants' nodding agreement*)

"... young people today, from the day they are born, have unlimited access to food ... they can go to the supermarket, they can eat snacks and chocolates every day, not only on Sundays or special occasions ... they have grown up without serious problems covering dietary and other basic, daily needs ..."

### 3.1.3. Residence Area (Rural/Urban)

In urban areas, the modern way of living entails a "distance" between nature and humans, involving limited available time for healthy eating habits, proper food-purchases, and cooking. Instead, in rural areas, there is still time, as well as respect for and awareness of nature. In addition, all the participants come from and/or have lived in a rural area (in the past). They have transferred their rural food-related habits and practices to urban living conditions. Some typical responses about the impact of residential place on HFW composition is presented below:

"... nowadays we can find whatever we want, all year round ... human needs are more demanding than ever and becoming more so"

"... we eat tomatoes only during spring and summer months ..."

"... young people are not close to nature anymore. When I was young, I used to work in our fields, in agriculture ... I know how and when food is produced ..."

"... I grew up in a rural area where we used to give food leftovers to our chickens and dogs. Living in the city now, I only feed birds on my balcony with stale bread ...".

### 3.1.4. Income and Education

Many of the responses indicated that income and education are two readily connected factors affecting HFW composition. Higher education enables individuals to achieve high

income levels; hence, high-income households are more likely to generate HFW. They can afford new food purchases without paying attention to cost and quantity. They are able to repurchase what they desire, covering their increasing food-related daily needs but within limited time constraints. The following response is an example of a high-income household attitude—a pharmacist working 6 days a week:

"… once a month I go to the supermarket. I have limited time as I have to return to my work … sometimes I order take out … I do not have enough energy to cook when I am back home …"

### 3.2. How Attitude Has an Impact on HFW

HFW composition is seriously affected by impulse food purchases (multipacks and discounts) and ignorance of food labeling dates—"expiry" and "best before". The available time for cooking has also negatively affected HFW composition. However, rural experiences and lifestyle contribute to a positive attitude preventing and/or reducing HFW composition. Furthermore, "Food Sharing" emerged as a particular habit widespread in rural Greece. Home-grown and/or traditional cooked food, e.g., homemade sweets, pies, etc. is shared among neighbors, friends, relatives, etc., usually using a household kitchen container. The container is not returned empty to the owners but is filled with home-grown or cooked food by the receivers. In the past, rural inhabitants used to promote whatever they produced in their "yard", selling or exchanging their produce in order to cover their essential daily nutritional needs. Since then, "Food Sharing" purposes and motivations have changed. Nowadays, it concerns social and interpersonal relationships preservation and/or development, based on ethical values, mostly among city dwellers who reminisce about the "rural ideal" [36]. Participants' most meaningful responses are listed below:

"… I am looking for discounts and/or multipacks … that help me save money …"

"… I feel safe with food labeling dates … we do not consume food after the "best before" date …"

"… I do not have enough time to cook every day … I cook enough twice a week and when someone feels hungry, he/she could find something to eat …"

"… every year my mother (rural inhabitant) makes a traditional red pepper and tomato sauce and gives me only 1–2 jars … since I do not have enough storage space in my apartment …"

"… in my rural residence we used to compost HFW in the courtyard … now we do not have enough space in our kitchen for a compost bin so we cannot to afford it …"

"… the town council has to place compost bins in every neighborhood across the city, we need them …"

"… I give eggs and vegetables from my garden to my daughter … my grandchildren love their quality and taste …"

### 3.3. How Knowledge and Perceptions concerning HFW Are Formed

The feedback also showed that the formed knowledge and perceptions concerning HFW have several impacts. Erroneous or incomplete perceptions were revealed. Participants cannot link their daily dietary and/or consumption choices with the global environmental issues (food miles, climate change, greenhouse effect, etc.). Additionally, the majority of responses revealed a weakness in understanding food labeling dates and the effectiveness of imposing fines on those who waste more than they should. However, economic crises had a positive influence on HFW composition, raising the willingness for prevention and reduction education. This is evidenced by the following responses:

"… I prefer local products … they are better quality than imported ones …"

"… I am not responsible for the greenhouse effect … the sun is …"

"… I am not responsible for the food waste worldwide … the supermarkets are … I have rejected food but not a large quantity … I am not in the habit of throwing away food …"

"… "expiry" and "best before" dates are the same information written in other words …"

"... "expiry" and "best before" dates may indicate something different about packaged food but certainly protect us from spoiled food ..."

"... during the economic crisis I bought the same domestic, quality agricultural products but in smaller quantities"

"... yes, I am interested to learn how to waste less ... it would be very interesting if someone could provide us with information on healthy nutrition and consumption habits ...".

## 4. Discussion

A plethora of studies are in progress on food wastage issues [2]. The investigation of food losses and waste in different countries around the world, focusing even on specific food types, e.g., tropical fruits [38], entails the necessity to organize and implement guidelines and strategies which will be consistent with sustainable nutrition and consumption [39]. Food wastage studies include economic, environmental, and social aspects which justify their multi-dimensional nature, emphasizing the urgency to examine each case in detail [40,41].

The economic costs at state, business, and household levels; the environmental degradation; and the nutritional-consumption inequalities are the main issues under exploration according to the FAO latest's works [42]. Consumption-households is that stage of the food supply chain where the largest quantities of waste have been recorded [2]. It is an imperative to prevent and/or reduce HFW, and the most indicated manner to achieve this goal is to investigate the factors affecting the HFW composition [43–45]. Therefore, the need for primary data is nowadays extremely important, especially for policy makers, food producers, retailers, and consumers—household members [40,46].

The present paper enriches the global bibliography as there is limited data on HFW in Greece, especially for rural areas [6,47,48]. More accurately, this study focuses on how and why the most common factors that make an impact on HFW composition [49,50] affect the household members using the case of Greece. Gender, age, education, income, and place of residence are the most common factors affecting HFW.

The research reveals a gender dimension in HFW composition. There is an unequal distribution of responsibilities and care initiatives between women and men, with regard to the consumption and eating habits in households. Women are mainly responsible for the daily food-related activities (shopping, cooking, cleaning, tidying, etc.) of their households. Consequently, women often find themselves rejecting and discarding food that was chosen or requested by other members of the household, rather than themselves. This is why the female gender has been directly linked to HFW production. Even in cases where these women lead busy lives, such as having professional obligations, they actively seek ways to manage household food sustainably. Moreover, they show a keen interest in nutrition and health, surpassing that of men. These factors collectively contribute positively to the prevention and/or reduction of HFW. Accordingly, the rural past and experiences have a positive influence on HFW composition. City dwellers who grew up and lived in rural areas have transferred environmentally friendly food-related habits to the urban households where they live now. In an urban environment, a socially constructed rurality has emerged, encompassing social, environmental, and economic interactions among household members. Fast-paced urban lifestyles, influenced by rural experiences, incorporate sustainable consumption and nutritional habits, resulting in reduced food waste and a greater respect for natural resources and farmers' efforts in production. For the first time in an HFW study in Greece, it is crucial to discuss the rural past and experiences.

In addition, two particularly important issues are identified through the current research. The "best before" date is confused with the "expiry" date, which is one of the most serious reasons that edible food ends up in household bins. This confusion is mainly found in food types such as legumes, pasta, cereals, and some dairy products. At the same time, the meaning of food miles, climate change, and greenhouse effect are not perceived, resulting in household members not being able to link their daily consumption and eating habits with global socioecological issues such as HFW. Household members are willing to be informed on the issues above

while local and governmental bodies ought to organize actions designed to provide accurate information, reducing the negative impact on HFW composition.

## 5. Conclusions

In the present study, an attempt was made to provide valuable information concerning HFW composition. An in-depth methodological approach was applied by collecting qualitative data on HFW factors, causes, and consequences. Two FGDs were conducted, developing a detailed interpretation of HFW in Greece, among rural and urban household members. The method used proved to be a valuable and flexible supplementary research tool in the field of rural sociology, including its application to the HFW issue. The FGDs methodology allows scientists to observe social interactions and explore individual knowledge, perceptions, and attitudes toward food-waste-related issues. In other words, it serves as an effective data collection method allowing scientists to gain insights into consumption and nutritional habits worldwide. This understanding could act as a specific strategic tool for planning and implementing policies, for example, informative campaigns, which could significantly contribute to HFW prevention and/or reduction. Thus, based on the consumption and nutritional specificities of each region, the desired results could first be achieved at the local level. The development of small-scale, local actions is capable of contributing to combating HFW across the country, thereby bringing us closer to achieving the goals of sustainable development.

Further qualitative studies based on the FGDs methodology should be attempted in the future. The innovative methodological tool used in this study offers an opportunity to all the involved parties to reveal special behavioral features and management methods on HFW issues, being customizable for particular geographical areas and/or customer groups.

## 6. Limitations

The limitations of this research pertain to the methodology, which specifically relies on qualitative research with a small sample size, where individual expressions prevail [36,51]. Furthermore, the results cannot be verified statistically, and they cannot be generalized [52,53]. This research offers findings that could inform future scientific studies. Moreover, the results encourage local authorities to develop strategies and implement informative actions, such as awareness campaigns and the placement of compost bins, to raise awareness of food waste issues.

**Author Contributions:** Conceptualization, V.A. and A.M.; methodology, V.A., A.M. and E.M.; validation, E.L. and G.T.; formal analysis, V.A., E.M. and A.M.; investigation, V.A., E.M. and G.T.; writing—original draft preparation, V.A., E.M. and A.M.; writing—review and editing, V.A., E.L., G.T., E.M. and A.M.; project administration, A.M. All authors have read and agreed to the published version of the manuscript.

**Funding:** This research received no external funding.

**Institutional Review Board Statement:** Ethical review and approval were waived for this study since the study was conducted in accordance with the Declaration of Helsinki and the EU General Data Protection Regulation.

**Informed Consent Statement:** Informed consent was obtained from all subjects involved in the study.

**Data Availability Statement:** Data will be available upon request by the first author.

**Conflicts of Interest:** The authors declare no conflicts of interest.

## Appendix A

**Table A1.** Focus group discussion guide.

| |
|---|
| 1. Who wastes the most food in your household? Men or women? Why? |
| 2. Young people and women are mainly interested in nutrient facts in food selection. Contrast, older people and rural residents are interested in qualitative characteristics (e.g., country of origin). Why?<br>3. The elderly waste less food than the young. Why?<br>4. Men and young people consume more meat and less fruits than women and older age people. Why?<br>5. How does income and educational level affect (positive/negative) HFW? Why? |
| 6. Do you use a food shopping list?<br>7. Why do not you compost food waste at home?<br>8. Do you waste food from vegetable gardens?<br>9. Do you consume food after the "use by" date?<br>10. How much time do you spend cooking/preparing food each day? Is it enough or not? Why?<br>11. Have you adopted any dietary/consumption habits stemming from your rural household/experience? |
| 12. What is the impact of economic crisis in household food waste management?<br>13. If fines, regarding food waste prevention/reduction are imposed, what would be fair (€/kg)?<br>14. How does food transportation and consumption affect the natural environment?<br>15. What is the connection between climate change, the greenhouse effect and HFW?<br>16. Is "use by" date the same as "expiry" date?<br>17. How could we reduce-prevent HFW through voluntary/informative actions? |

**Table A2.** Answer sheet.

| Date | |
|---|---|
| Age | |
| Residence | Urban/Rural |
| Education | Primary/Secondary/Post-secondary/Tertiary |
| Questions | Answers |
| 1 | |
| 2 | |
| 3 | |
| 4 | |
| 5 | |
| 6 | |
| 7 | |
| 8 | |
| 9 | |
| 10 | |
| 11 | |
| 12 | |
| 13 | |
| 14 | |
| 15 | |
| 16 | |
| 17 | |

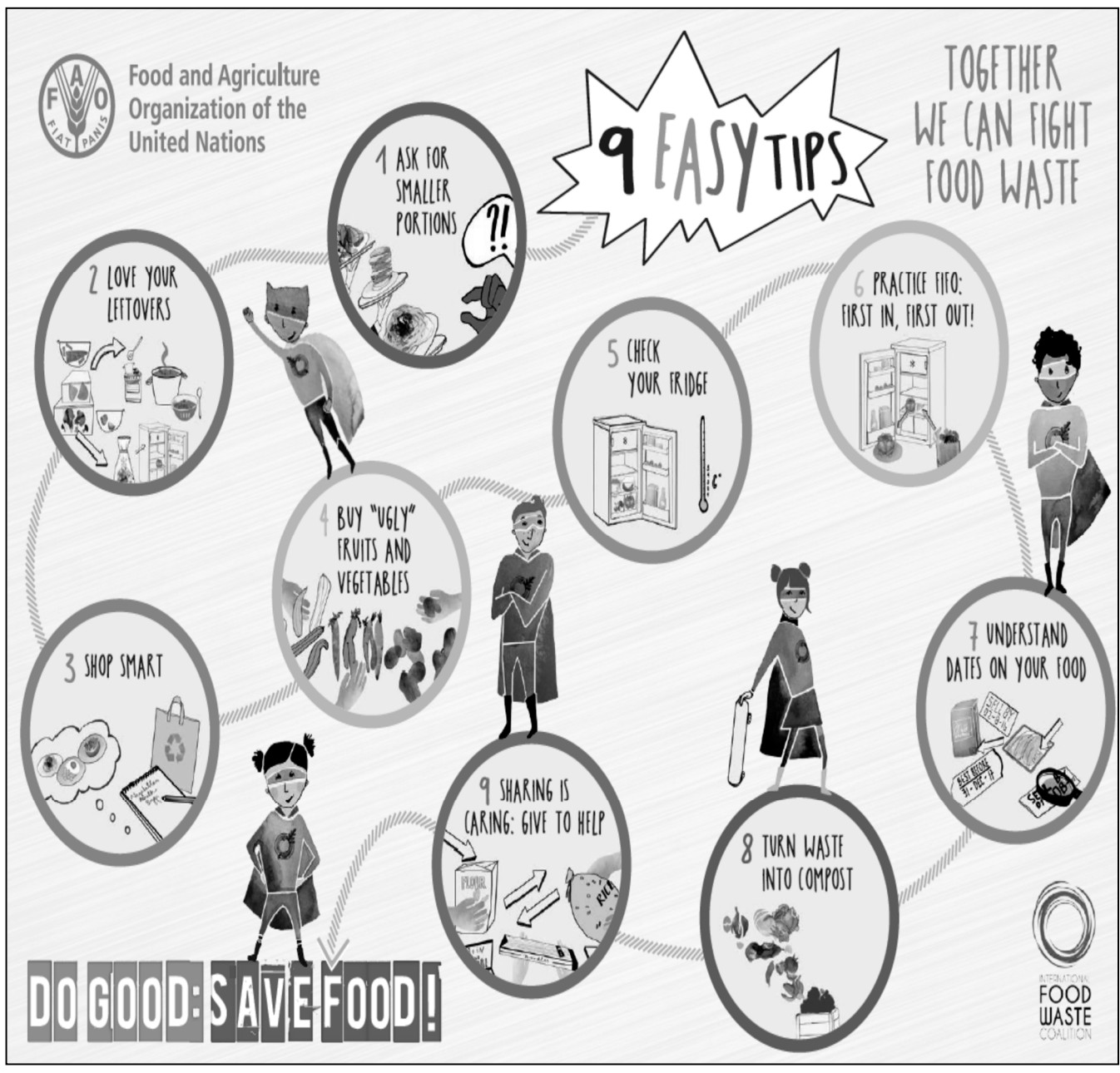

**Figure A1.** Practical advice about how to reduce HFW [33].

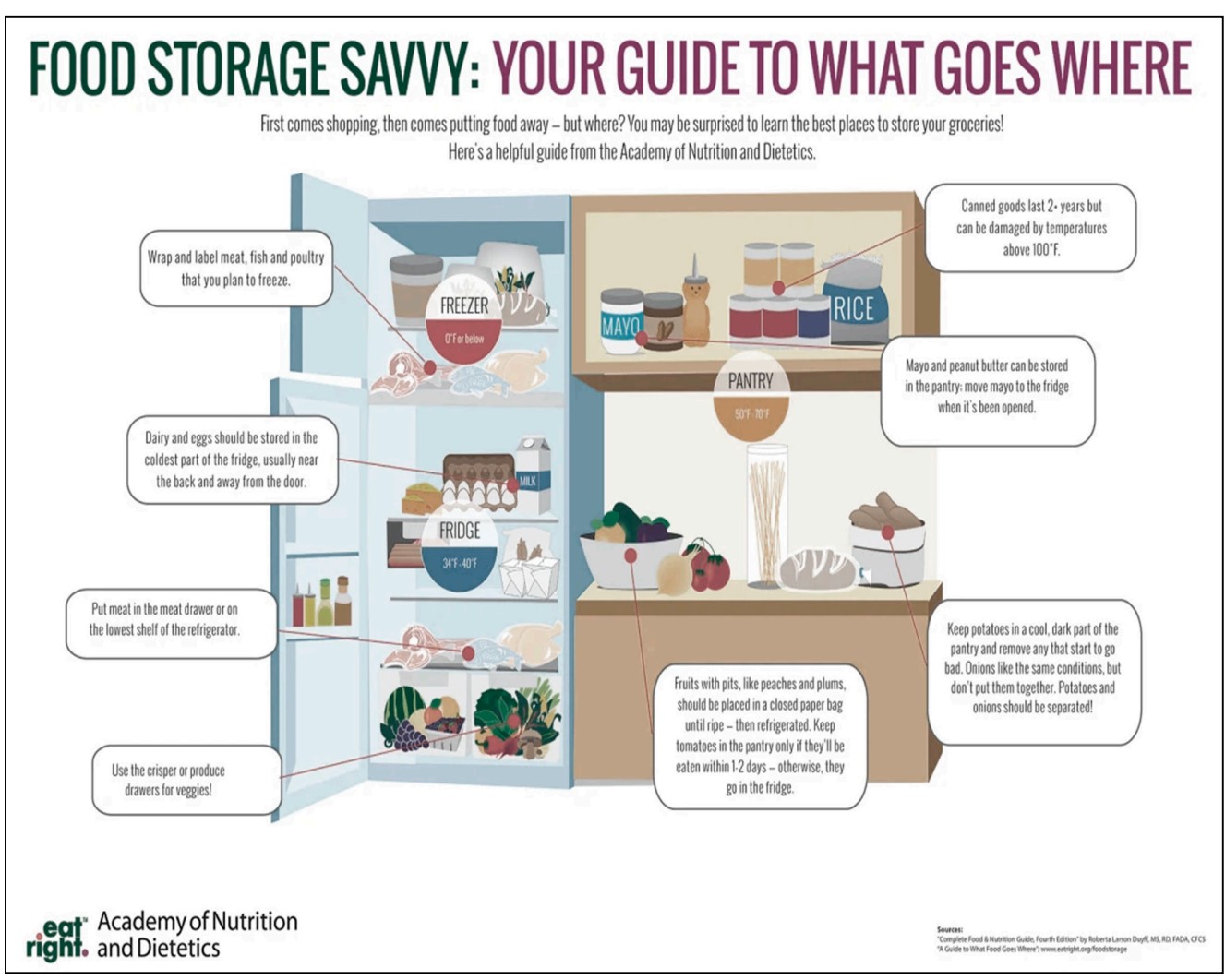

**Figure A2.** "Every Food on the Right Shelf" [34].

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
