# Peer review of "Focus Group Discussions on Food Waste: An Empirical Application Providing Insights into Rural and Urban Households in Greece"

_sustainability, doi:10.3390/su16020502_

Round 1

Reviewer 1 Report

Comments and Suggestions for Authors

All comments and corrections are incorporated in the pdf version.

Comments on the Quality of English Language

The quality of the English language is at a satisfactory level. All corrections have been incorporated into the text

Author Response

 listing the reviewer comments and our specific response to those comments

Reviewer 1 Comments

Response

1

Check the number of words in the abstract. According to the instructions, the abstract should contain up to 200 words

Has been done.

The abstract contains 212 words.

2

References must be numbered in order of appearance in the text

Has been done.

All the references have been numbered in order of appearance in the text.

3

Delete or rephrase this paragraph

Has been done.

The paragraph has been rephrased, presenting information about the paper’s context and its innovation on the scientific field of Food Waste. – lines 97 - 104

4

Please add a sentence stating that the following content consists of the respondents' statements or answers.

– line 275

Has been done.

A sentence stating that the following content consists of the participant’s responses has been added: “This is evidenced by the following responses:” – line 304

5

Please separate Discussion from Conclusions.

Has been done.

Discussion and Conclusions has been separated. In the Discussion section has been added 2 more paragraphs – lines 320- 334.

6

Grammatical and vocabulary suggestions.

Has been done.

All the suggestions have been added in the text.

7

Format references in accordance with the instructions for authors

Has been done.

All the references have been formatted according to the journal’s instructions.

Reviewer 2 Report

Comments and Suggestions for Authors

My comments are as follow:

·         In the keyword section needs to be improved, since some words of the manuscript´s title are repeated, I suggest to replace the words: “focus group”; “food waste”; “rural households”; “women's food habits” by other keywords to improve the scope of your research in the scientific websites databases.

·         The conclusions – discussions and limitations section are very extensive. I suggest dividing all those sections, and in this way improving each one, independently.

I am attaching my answers to your queries, that are shown in blue font: 

1. What is the main question addressed by the research?

The main question was “This research addresses seven crucial findings regarding HFW composition providing a better understanding of them”. Please, you will find that information in lines from 66 to 77.

2. Do you consider the topic original or relevant in the field? Does it
address a specific gap in the field?

In my point of view, the specific gap in the field is the “Focus Group Discussions on Food Waste: An Empirical Application Providing Insights into Rural and Urban Households in Greece”.

3. What does it add to the subject area compared with other published
material?

In my point of view, the add to the subject area compared with other published material is the “Focus Group Discussions on Food Waste: An Empirical Application Providing Insights into Rural and Urban Households in Greece”.

4. What specific improvements should the authors consider regarding the
methodology? What further controls should be considered?

•             In the keyword section needs to be improved, since some words of the manuscript´s title are repeated, I suggest to replace the words: “focus group”; “food waste”; “rural households”; “women's food habits” by other keywords to improve the scope of your research in the scientific websites databases.

•             It is very important to include more precise statistical analyses of the data obtained from the research.

•             The conclusions – discussions and limitations section are very extensive. I suggest dividing all those sections, and in this way improving each one, independently.

5. Are the conclusions consistent with the evidence and arguments presented
and do they address the main question posed?

In my point of view, are very basic for a “Original Article”. The authors must correct the conclusion section, according to the objectives of the research.

6. Are the references appropriate?

The authors of the manuscript need to improve the references.

7. Please include any additional comments on the tables and figures.

It is mandatory to include more accurate statistical analysis of the research data, especially in tables and figures of the manuscript.

Author Response

 listing the reviewer comments and our specific response to those comments

Reviewer 2 Comments

Response

1

In the keyword section needs to be improved, since some words of the manuscript´s title are repeated, I suggest to replace the words: “focus group”; “food waste”; “rural households”; “women's food habits” by other keywords to improve the scope of your research in the scientific websites databases.

Has been done.

All the keywords have been replaced, as follows: factors, household food waste, interviews, qualitative research, rurality

2

It is very important to include more precise statistical analyses of the data obtained from the research.

Has been done.

More precise statistical data, of the empirical research, have been added in order to provide a clearer understanding on the necessity to implement the supplementary research of FGDs methodology – lines 66-96.

3

The conclusions – discussions and limitations section are very extensive. I suggest dividing all those sections, and in this way improving each one, independently

Has been done.

Discussion, conclusions and limitations have been separated in order to provide a better understanding on each section. More specifically the Discussion section has been enriched by more references – lines 320-334.

4

The authors of the manuscript need to improve the references.

Has been done.

All the references have been numbered in order of appearance in the text and have been formatted according to the journal’s instructions.

5

It is mandatory to include more accurate statistical analysis of the research data, especially in tables and figures of the manuscript

Has been done.

The current study presents a supplementary qualitative research methodology on HFW in order to provide clear evidence on the statistical findings of the large-scale empirical research. The necessary statistics have been added in the text – lines 66-96.

Reviewer 3 Report

Comments and Suggestions for Authors

The manuscript is good, although it should be expanded especially in terms of conclusions, implications and future research. I congratulate the authors for it! However, in this form, I believe that it cannot be published.

The comments and recommendations for publication are:

1. The authors mentioned that: "The data for this research were collected in a sample of Greek population". How was the size of this sample determined for it to be relevant? How were the respondents identified so that their answers are relevant? If there was no criterion for both sample size and respondent identification, this must be mentioned. Depending on these, the work can be reviewed in one way or another. It is the most problematic aspect of the manuscript.

2. The attached figures can be interpreted by the authors in a personal way. It would bring more value and authenticity to the work.

3. Bibliographic references must be completed with eloquent data. Even one of the first author's articles did not mention the DOI (I found it and have to include DOI: 10.1108/BFJ-02-2019-0111). In addition, the list of bibliographic references should be expanded to give credibility to the research. With certainty 37 titles do not represent a credible list to support the presented research.There is a huge literature in the field. Some suggestions:

Durán-Sandoval, D.; Durán-Romero, G.; Uleri, F. How Much Food Loss and Waste Do Countries with Problems with Food Security Generate? Agriculture 2023, 13, 966. https://doi.org/10.3390/agriculture13050966

Tansuchat, R.; Pankasemsuk, T.; Panmanee, C.; Rattanasamakarn, T.; Palason, K. Analyzing Food Loss in the Fresh Longan Supply Chain: Evidence from Field Survey Measurements. Agriculture 202313, 1951. https://doi.org/10.3390/agriculture13101951

Shafiee-Jood, Majid, and Ximing Cai. "Reducing food loss and waste to enhance food security and environmental sustainability." Environmental science & technology 50, no. 16 (2016): 8432-8443https://doi.org/10.1021/acs.est.6b01993

Serafini, M.; Toti, E. Unsustainability of Obesity: Metabolic Food Waste. 
Front. Nutr. 2016, 3, 40. Available online: https://www.frontiersin.org/article/10.3389/fnut.2016.00040 

Toti, E.; Di Mattia, C.; Serafini, M. Metabolic Food Waste and Ecological Impact of Obesity in FAO World’s Region. Front. Nutr. 2019, 6, 126. Available online: https://www.frontiersin.org/article/10.3389/fnut.2019.00126 

Romani Simona, Silvia Grappi, Richard P. Bagozzi, Ada Maria BaroneDomestic food practices: A study of food management behaviors and the role of food preparation planning in reducing waste, Appetite, Volume 121, 2018, Pages 215-227, ISSN 0195-6663, https://doi.org/10.1016/j.appet.2017.11.093.https://www.sciencedirect.com/science/article/pii/S0195666317308516

4. Citing works as bibliographic references in a language other than English (in Greek in the present case, respectively the doctoral thesis of the first author of this article) is not at all effective in order to be able to review the manuscript. Maybe the MDPI editors allow such a thing, but categorically these references can only be accessed by the majority of reviewers, who do not know this language, which is also my case.

Author Response

 listing the reviewer comments and our specific response to those comments

Reviewer 3 Comments

Response

1

How was the size of this sample determined for it to be relevant? How were the respondents identified so that their answers are relevant? If there was no criterion for both sample size and respondent identification, this must be mentioned.

Has been done.

The paper includes information regards to the sample size of the empirical research, sample’s technique and statistical analyses which led the researchers to implement the qualitative research methodology of FGDs in order to provide a clearer insight on HFW causes and consequences – lines 106-115.

2

Can be an original interpretation of it – Figure 1 & 2

Has been done.

In lines 150 – 158 there is an explanation about the content of each figure and their purposes during the research process.

3

This is a thesis in Greek. Not readable for reviewers.

Has been done.

The reference of the Greek thesis has been deleted.

4

Have to include

DOI: 10.1108/BFJ-02-2019-0111

Has been done.

The DOI number has been added – line 428.

5

Bibliographic references must be completed with eloquent data. Even one of the first author's articles did not mention the DOI (I found it and have to include DOI: 10.1108/BFJ-02-2019-0111). In addition, the list of bibliographic references should be expanded to give credibility to the research. With certainty 37 titles do not represent a credible list to support the presented research. There is a huge literature in the field. Some suggestions:

Has been done.

The references have been improved significantly. More specifically, all the references have been numbered in order of appearance in the text and have been formatted according to the journal’s instructions. Also, the reference list has been expanded, giving credibility to the research, by 16 more reports – lines 477-507.

6

Citing works as bibliographic references in a language other than English (in Greek in the present case, respectively the doctoral thesis of the first author of this article) is not at all effective in order to be able to review the manuscript. Maybe the MDPI editors allow such a thing, but categorically these references can only be accessed by the majority of reviewers, who do not know this language, which is also my case.

Has been done.

The Greek thesis reference has been deleted.

Reviewer 4 Report

Comments and Suggestions for Authors

Thank you for giving me the opportunity to revise the MS entitled “Focus Group Discussions on Food Waste: An Empirical Application Providing Insights into Rural and Urban Households in Greece” by Aitsidou and his/her colleagues that was submitted to “sustainability”.  This manuscript lacks innovation, the text is chaotic. I don't think it meets the requirements of the journal. The author needs to carefully consider the innovation of this manuscript and carefully revise it.

Author Response

 listing the reviewer comments and our specific response to those comments

Reviewer 4 Comments

Response

1

This manuscript lacks innovation, the text is chaotic. I don't think it meets the requirements of the journal. The author needs to carefully consider the innovation of this manuscript and carefully revise it.

Has been done.

The manuscript has been revised. More specifically, have done the changes and improvements below:

1.       All the keywords have been replaced, as follows: factors, household food waste, interviews, qualitative research, rurality, in order to improve the scope of the research in the scientific websites databases.

2

2.       More precise statistical data, of the empirical research, have been added in order to provide a clearer understanding on the necessity to implement the supplementary research of FGDs methodology – lines 66-96.

3.       The paper includes information regards to the sample size of the empirical research, sample’s technique and statistical analyses which led the researchers to implement the qualitative research methodology of FGDs in order to provide a clearer insight on HFW causes and consequences – lines 106-115.

3

4.       Discussion, conclusions and limitations have been separated in order to provide a better understanding on each section. More specifically the Discussion section has been enriched by more references – lines 320-334. In the section of conclusions has been added a paragraph focusing on study’s innovation and its impact for future research.

4

5.       The references have been improved significantly. More specifically, all the references have been numbered in order of appearance in the text and have been formatted according to the journal’s instructions. Also, the reference list has been expanded, giving credibility to the research, by 16 more reports – lines 477-507.

5

6.       A paragraph has been rephrased, presenting information about the paper’s context and its innovation on the scientific field of Food Waste. – lines 97 – 104. Also, the innovation of the current study is provided in conclusions – lines 372-391.

6

7.       Rephrases and vocabulary changes have been done throughout the text.  

Round 2

Reviewer 2 Report

Comments and Suggestions for Authors

Accepted after minor revision.

Reviewer 3 Report

Comments and Suggestions for Authors

The manuscript has been completed and improved to meet publication requirements.